# The Need for Red Cell Support During Non-Cardiac Surgery Is Associated to Pre-Transfusion Levels of FXIII and the Platelet Count

**DOI:** 10.3390/jcm9082456

**Published:** 2020-07-31

**Authors:** Silke Listyo, Eric Forrest, Lukas Graf, Wolfgang Korte

**Affiliations:** 1University of Bern, Hochschulstrasse 6, 3012 Bern, Switzerland; silke.listyo@usz.ch; 2Blutspendezentrum SRK Ostschweiz, Rorschacher Strasse 111, 9000 St. Gallen, Switzerland; eric.forrest@blutspende-sg.ch; 3Center for Laboratory Medicine and Haemostasis and Hemophilia Center St. Gallen, Frohbergstrasse 3, 9001 St. Gallen, Switzerland; lukas.graf@zlmsg.ch

**Keywords:** intraoperative bleeding, pre-transfusion level, FXIII activity, platelet count, fibrinogen, transfusion, red cell support

## Abstract

Unexpected intraoperative bleeding is associated with a reduced availability of crosslinking capacity (provided through factor XIII (FXIII)) per unit of generated thrombin. Furthermore, FXIII deficiency and thrombocytopenia (but not fibrinogen deficiency) are the most prevalent modulators of clot firmness in the immediate postoperative setting. In this study, we therefore evaluated whether levels of FXIII, fibrinogen, or the platelet count influenced the probability of intraoperative red cell transfusions in patients in the operating theatre. This retrospective study was comprised of 1023 patients, which were in need of blood product support in the operating theatre and of which 443 received red cell transfusions. Due to standard operating procedures, FXIII activity, fibrinogen concentration, and platelet count were measured before transfusion took place, but without influencing the decision to transfuse. FXIII deficiency was frequent (50%), as was thrombocytopenia (49%), but not fibrinogen deficiency (9%). FXIII deficiency was associated with a significantly increased probability to receive red cell transfusions (OR 4.58, 95% CI 3.46–6.05) as was thrombocytopenia (OR 1.94, 95% CI 1.47–2.56), but not fibrinogen deficiency (OR 1.09, 95% CI 0.67–1.76). Similar results were seen for cut-off independent evaluations (receiver operating characteristics (ROC) curves, using continuously distributed variables), where the areas under the curves (AUC) of red cell transfusion for FXIII activity was 0.744 (95% CI 0.716–0.770)/0.632 (95% CI 0.601–0.661) for the platelet count, and 0.578 (95% CI 0.547–0.609) for fibrinogen concentration. All AUCs were significantly different from each other (*p* < 0.0001 and *p* = 0.0106, respectively), indicating that FXIII activity was a significantly better predictor of red blood cell (RBC) transfusion than platelet count and fibrinogen concentration. These results suggest that pre-transfusion FXIII activity and to a lesser extent the platelet count influence the probability of intraoperative red cell transfusions. Modifying FXIII activity and/or the platelet count might influence the need for downstream red cell transfusion, thus potentially reducing transfusion associated morbidity. This, however, needs confirmation in future studies.

## 1. Introduction

Unexpected and unexplained intraoperative coagulopathies and bleeding continue to be a diagnostic and therapeutic dilemma. Peri- and intraoperative blood loss may lead to acute anemia that, depending on its extent, might be life threatening. This is due to the fact that compensatory mechanisms, unlike in chronic anemia, had no time to be established at that point. Therefore, perioperative blood loss, if leading to hemodynamic instability and/or organ dysfunction, might require red blood cell (RBC) transfusion. RBC transfusions, however, have been shown to be associated to relevant morbidity [1,2,3] and even mortality [2,4]. Various reasons exist for the explanation of these phenomena. Most frequently, adverse events leading to fatalities in the context of RBC transfusion are transfusion-related acute lung injury (TRALI), hemolytic transfusion reactions, circulatory overload, and infections due to known pathogens. Other, not yet well defined risks are derived from possible immunomodulation, unknown emerging infectious agents, and possible storage lesions [3]. In order to try to prevent such negative impacts, alternative management scenarios other than RBC transfusions are necessary. For this reason, it would be desirable to be able to identify patients upfront who have an increased risk for intraoperative transfusion requirements. If a modifiable pathophysiologic mechanism leading to increased transfusion requirements could be identified, this might allow early therapeutic or even prophylactic interventions to reduce morbidity and/or mortality associated with bleeding and RBC transfusions. Multiple factors can impair the coagulation system and may therefore lead to bleeding and/or major blood loss in the surgical setting. The type of surgery, preexisting comorbidities, currently used medication, intra-operatively used drugs, and congenital or acquired coagulation disorders are some important factors [5]. Specifically, patients with preoperative anemia [6,7,8], elderly patients [1], and females [9] have been shown to be at higher risk for perioperative RBC transfusion. Recent work of various clinical research groups has helped us to better understand the potential mechanisms that can explain the pathophysiology of so far unexpected and unexplained intraoperative bleeding and blood loss. There is growing evidence that clotting factor XIII (FXIII) plays a major role in this context. Circulating FXIII is a tetramer consisting of two identical proenzymes (FXIII-A2) and two carrier subunits (FXIII-B2). The active part, FXIII-A, is a transglutaminase. It contains an NH_2_-terminal activation peptide and Ca^2+^-binding sites on its surface. These two structural elements are highly relevant for the activation process of FXIII, since thrombin cleaves the activation peptide FXIII from the NH_2_-Terminus of FXIII-A, while binding of Ca^2+^ to the binding sites leads to dissociation of the inhibitory FXIII-B2. This lets FXIII(-A) merge into its active conformation, allowing it to mount its cross-linking capacity. Soluble fibrin-monomers, fibrin and α_2_-plasmin inhibitors are the main substrates of activated FXIII. Activation of FXIII triggers its major task, cross-linking soluble fibrin monomers generated through liberation of fibrinopeptides from fibrinogen under the influence of thrombin [10,11,12]. Under these conditions, a stable fibrin network as a basis for an appropriately firm clot and its resistance towards fibrinolysis can develop.

FXIII deficiency can be either congenital or acquired. Congenital FXIII deficiency is a rare, autosomal recessively inherited condition with an estimated frequency of one in 2 to 3 million live births. It is usually based on a mutation in the FXIII-A subunit (95% of all FXIII deficiencies). Mutations in the FXIII-B subunit are rare. Congenital FXIII deficiency accounts for roughly 6% of rare bleeding disorders [13]. The reasons for the development of acquired FXIII deficiency are manifold. Three different main groups can be defined based on the underlying mechanisms and/or disease groups: (1) autoimmune conditions (e.g., systemic lupus erythematosus; rheumatoid arthritis; malignancy; medications (e.g., isoniazid); monoclonal gammopathy of undetermined significance); (2) increased consumption (e.g., surgery; disseminated intravascular consumption (DIC), including sepsis; inflammatory bowel disease; Henoch–Schönlein purpura; Thrombosis); and (3) decreased synthesis (e.g., liver disease; leukemia; medication (e.g., valproic acid, tocilizumab)) [14]. A very rare entity of acquired FXIII deficiency is a hemophilia-like disease caused by an anti-FXIII antibody, leading to potentially life threatening bleeding events [15]. By far the most prevalent reason for acquired FXIII deficiency is consumption in the surgical setting [16]. Treatment therefore mostly consists of FXIII substitution, sometimes in combination with immunosuppressive therapy if an autoimmune or specific inhibitory component is present. In conclusion of these considerations, FXIII activity depends on many variables. Since surgery itself leads to consumption of FXIII, the prevalence of FXIII deficiency in surgical patients can be expected to be high. This might be exacerbated in the perioperative setting with the combination of various and additional risk factors in the same patient.

## 2. Background

Our group has previously shown that patients undergoing elective surgery (and in which unexplained and unexpected intraoperative bleeding occurs) exhibit a higher degree of prothrombin conversion as well as reduced availability of crosslinking capacity (most likely due to reduced FXIII activity) per unit of generated thrombin. This finally leads to reduced clot firmness and higher blood loss [17]. In line with these findings, we demonstrated that fibrin monomer concentrations in patients with intraoperative blood loss due to coagulopathy, quantified through immunological assays, are persistently increased in the pre-, peri-, and postoperative period, and can in fact be used for preoperative risk stratification [18]. Reduced fibrin monomer crosslinking due to reduced FXIII availability is also able to explain the persistence of increased fibrin monomer concentration with ongoing thrombin generation. Other groups therefore also hypothesized that FXIII plays a key role in perioperative bleeding [19,20,21]. To test this hypothesis, we performed a proof of principle study using FXIII early during surgery in high-risk patients [22]. The use of FXIII led to a significant reduction of fibrinogen consumption, blood loss, and loss of clot firmness [22,23]. In the immediate postoperative setting, clot firmness, as determined by thrombelastography, has been associated to the platelet count and the fibrinogen level [24]. On top of these observations, we identified FXIII as an independent predictor of overall clot firmness by multivariate analysis [16], again including platelet count and fibrinogen In that study, FXIII deficiency was very frequent immediately after surgery, as was thrombocytopenia. Fibrinogen deficiency was rather rare. Comparable observations were also reported by other groups [25,26,27,28]. It seems therefore reasonable to believe that the conditions found very early postoperatively might be similar to the ones present during surgery.

We were therefore interested to find out whether the modulators of clot firmness we had identified in the immediate postoperative phase might also be determinants of the future need of intraoperative RBC support. Using a look back analysis, we evaluated the potential association of pre-transfusion FXIII activity, fibrinogen level, and platelet count with intraoperative red cell transfusion in a routine surgical setting.

## 3. Patients and Methods

A two-year look back analysis was used to identify patients who had experienced intraoperative bleeding and for whom blood products (fresh frozen plasma (FFP), red cell concentrates, or platelet concentrates) were ordered to the operating theatre. From this delivery list of the Blood Center, orders for RBC concentrates were identified. Patients were from the general surgery, orthopedic surgery, hand surgery, and the urology department. The study was approved by the regional review board (EKOS, ID 2020-01358). At the Kantonsspital St. Gallen, a regional tertiary care center, the strategy is to transfuse only if the clinical situation of the patient with regard to hemodynamic stability and/or long term outcome indeed requires it, as well as if transfusing blood products can be considered safe. The final decision on whether or not to transfuse is at the discretion of the senior anesthesiologist in charge. As a standard of care procedure in patients with planned use of blood products, all patients had routine platelet count, fibrinogen, and FXIII determination performed immediately before red cell transfusion was given. This was done to document the baseline values of the respective components before transfusion therapy was started. In order to avoid treatment (i.e., transfusion therapy)-associated bias, data from repeat blood product deliveries for the same patient were excluded. Thus, only the first RBC transfusions during a given hospitalization were evaluated. A detailed overview of the patient cohort including pre-transfusion levels of FXIII, platelet count and fibrinogen as well as the number of RBCs transfused is provided as Appendix A. Coagulation factor deficiencies or thrombocytopenia were defined as results below the lower border of the respective reference range (150 G/L for the platelet count, 1.5 g/L for fibrinogen concentration and 70% for FXIII activity). Data were evaluated in three ways: first, the prevalence of the respective coagulation factor deficiencies alone or in combination were determined; second, odds ratios for the transfusion of red cell concentrates in the whole cohort (i.e., comparing patients with RBC transfusion to those without RBC transfusions) depending on whether or not a deficiency (or a combination of deficiencies) was present, were calculated; third, from continuously distributed data, receiver operating characteristics (ROC) curves were constructed on the use of red cell concentrates for every variable; and ROC curves’ areas under the curves (AUC) were compared to the number of red cell concentrates transfused if one abnormality was present (Mann–Whitney testing). As blood draws were routine draws, they were performed according to standard operating procedures (Becton Dickinson Vacutainer^®^ containing 0.129 M tri-sodium-citrate (coagulation assays) or Becton Dickinson K_2_EDTA Vacutainer^®^ (platelet count) were used; (both from Becton Dickinson AG, Basel, Switzerland)). Platelet counts were measured on a Sysmex XE analyzer (Sysmex Europe GmbH, Horgen, Switzerland). FXIII activity was determined using Berichrome assay on a BCS analyzer (both from Siemens, Marburg, Germany). Fibrinogen concentrations were measured according to the Clauss method on an ACL Top analyzer (Instrumentation Laboratory, Milan, Italy). All assays were performed according to the recommendations of the manufacturers. All statistical evaluation was done using MedCalc Software, version 19.4.0 (MedCalc Software, Ostend, Belgium) [29].

## 4. Results

After excluding repeat deliveries, data sets of 1023 patients who had blood products delivered to the operating theatre, were identified for further evaluation. Of those, 443 had first time deliveries and transfusions of RBCs. A total of 1636 red cell concentrates were transfused in these 443 patients. Determinations from blood draws in the operating room (all 1023 patients) immediately before transfusions showed an overall median platelet count of 151 G/L (95% CI 145–157 G/L; reference range 150–300 G/L), median fibrinogen concentration of 2.53 g/L (95% CI 2.47–2.61 g/L; reference range 1.5–3.5 g/L), and median FXIII activity of 69% (95% CI 66–73%; reference range 70–140%). Overall, FXIII deficiency (<70%) was very frequent and the most prevalent abnormality observed (50% of all observations), with thrombocytopenia (<150 G/L) being almost as frequent (49%); in contrast, fibrinogen deficiency (<1.5 g/L) was rare (9%, see Table 1). Table 1 also includes the frequencies for the respective deficiencies if lower cut-off levels than regular reference range borders were used. Not unexpectedly, these results closely match ones in the early postoperative setting [16].

Given the results observed in our studies so far, it seems likely that combinations of deficiencies of pro-coagulant factors were important for clinical outcome. Therefore, we additionally evaluated the prevalence of various combinations of deficiencies for FXIII, fibrinogen, and thrombocytopenia (Table 2). It can be seen that FXIII deficiency (rows B, C, E, F; 50%) and thrombocytopenia (rows B, D, E, G; 49%) alone or in combination show a similarly high prevalence while fibrinogen deficiency alone or in combination has a low prevalence (rows E, F, G, H; 9%).

In our cohort of 1023 patients, a FXIII deficiency of <70% was associated with a 4.6-fold risk of receiving red cell concentrates, while this risk was approximately twofold for patients with thrombocytopenia. Interestingly, there was no increased probability for patients with fibrinogen levels < 1.5 g/L (Table 3). When combined deficiencies were evaluated, only the combination of FXIII deficiency and thrombocytopenia were associated with a significantly increased risk to receive RBCs; the combination of FXIII deficiency and thrombocytopenia was not significantly different from the risk associated to FXIII or thrombocytopenia alone (Table 3).

When ROC curves were constructed (i.e., if continuously distributed data rather than categorized data were used) (Figure 1), it can be seen that that all three variables are (statistically speaking) potential predictors for the use of red cell concentrates (95% CI for the area under the curve for all variables significantly different from 0.5, Table 4).

The respective AUCs (see Table 4) of the various ROC curves were significantly different from each other (*p* < 0.0001 and *p* = 0.016, respectively). This indicates that pre-transfusion FXIII activity was a significantly better predictor of RBC transfusion than platelet count and fibrinogen concentration in this cohort study (Table 4).

## 5. Discussion

To date, perioperative blood loss and the need for intraoperative transfusion cannot be reliably predicted by preoperative laboratory parameters. For now, single routine laboratory assays fail to predict relevant perioperative coagulopathy and/or transfusion needs and are therefore not useful for preoperative risk stratification [30]. Preoperative risk assessment is frequently attempted by using standardized questionnaires on bleeding and medication history (e.g., use of antiplatelet drugs) [31]. Outside the use of antithrombotics, however, this neither allows for the identification of a bleeding risk (occurring unexpectedly) during surgery nor the identification of a reason for that bleeding risk. Both would be necessary in order to counteract a potential bleeding risk with a remedy based on the respective pathophysiology. Treatment of the underlying pathophysiology must be the ultimate goal.

Nevertheless, there is evidence that patients at risk for intraoperative bleeding are accessible to preoperative risk stratification with some biomarkers [18], particularly increased fibrin monomer concentration determined in reducing conditions. We were able to demonstrate in a prospective, controlled study (outside cardiac surgery) that fibrin monomer allows preoperative risk stratification for increased intraoperative blood loss [32]. Exploring the pathophysiology behind these findings, we have shown that patients developing an unexplained intraoperative bleeding event during non-cardiac surgery show decreased clot firmness early during surgery. This reduction in clot firmness was found to be secondary to reduced cross linking capacity, associated with the reduced availability of FXIII per unit of generated thrombin [17]. We suggest that the aforementioned increase in fibrin monomer concentration is a sign of an imbalance between fibrin monomer generation (through the influence of thrombin) and the FXIII activity, which is available in reduced quantity. This decreases the conversion of (measurable) soluble fibrin monomers into a non-soluble (thus no longer measurable), cross-linked fibrin network. This, accordingly, will increase soluble fibrin monomer concentrations in patients with decreased FXIII activity.

In line with this theory, increasing the availability of FXIII in patients at risk (through the early infusion of FXIII concentrate) significantly reduces loss of clot firmness, loss, and/or consumption of fibrinogen and blood loss, as evidenced in a proof of principle, randomized, double-blind, placebo controlled trial [22]. Furthermore, we showed that early postoperative clot firmness is mainly dependent on FXIII, platelet count, and fibrinogen, with FXIII deficiency and thrombocytopenia being very prevalent features [16]. Most recently, FXIII is the only consistent antepartum predictor of increased postpartum blood loss in patients with postpartum hemorrhage [33]. Other investigators have found similar corresponding results when evaluating risk factors for postoperative hematoma in neurosurgical patients [21]. Our present cohort study suggests that reduced FXIII, in line with our earlier findings on blood loss and clot firmness, significantly influences the need of intraoperative red blood cell transfusion. It is important to reiterate that the indication for RBC transfusion in our center neither depended on FXIII activity or fibrinogen concentration nor on the platelet count, but on the clinical picture and the hemoglobin concentration. As blood draws for laboratory measurements were performed only after the decision to order blood products was made, we cannot exclude the possibility that some consumption had already taken place at this point in time. As only data of first time orders were taken into account in our cohort, it is very unlikely though not impossible that any of the results of the coagulation data influenced the decision to transfuse red cell concentrates.

Further studies will thus need to evaluate whether modifying coagulation factors (e.g., FXIII, platelet count) can reduce the need for red cell transfusions in patients at risk in non-cardiac surgery; and whether not only pre-transfusion but also preoperative FXIII values might be predictive of future transfusion needs. Earlier studies of FXIII replacement in cardiac surgery yielded different results, possibly due to various methodological issues and varying definitions of patients at risk, i.e., patient selection [19,34]. Although the results presented here suggest a strong association between FXIII deficiency and, to a lesser extent, thrombocytopenia and the need for RBC support, the retrospective nature of our study hinders definitive proof of this concept. Furthermore, the single center character of our study entails the risk of a potential local bias. However, the endpoint of our study is concise (i.e., whether or not RBC concentrates were transfused). In addition, our data base is very large and analysis was done under application of stringent criteria. The results presented here are in line with findings from our own and from other groups. We therefore believe the results provide sufficient evidence to allow generation of the hypothesis described, i.e., that modulation of FXIII activity and /or the platelet count might influence the need for RBC transfusions.

## 6. Conclusions

In line with earlier findings by our group and others, we herein showed that FXIII and the platelet count were significant predictors of transfusion of RBC concentrates in a large cohort of various visceral, non-cardiac surgery patients. Results for fibrinogen as a predictor are not homogenous, showing a very small predictive value in ROC curve analysis (but significantly less than the platelet count or FXIII), but not in multivariate analysis. To minimize or prevent the need for intraoperative blood product support, strategies that allow preoperative risk stratification for intraoperative bleeding risk, including potential biomarkers, are urgently needed. As the early use of FXIII in high risk surgical patients resulted in reduced blood loss, one can assume that it will also result in reduced transfusion requirements [22]. However, in order to confirm the relevance of the results found here for intraoperative transfusion management, prospective intervention studies with the early use of FXIII and/or platelet transfusions in patients at risk are needed.

## Figures and Tables

**Figure 1 jcm-09-02456-f001:**
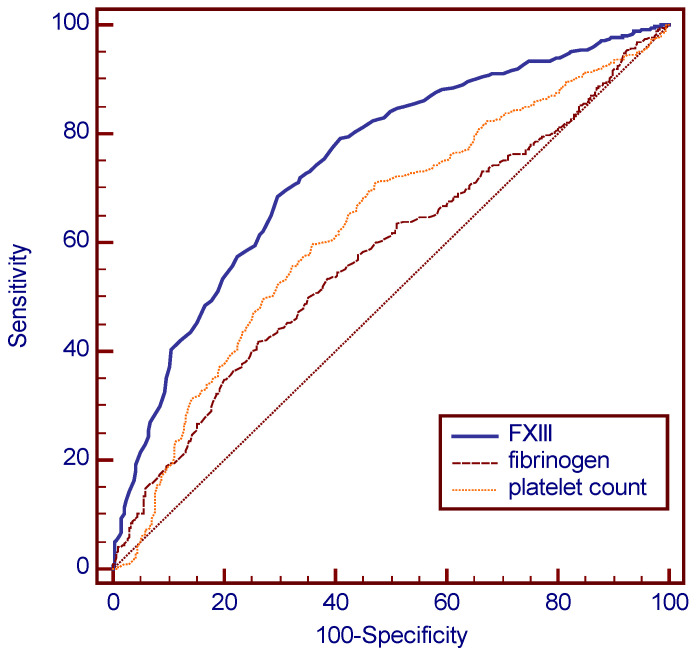
Comparison of ROC curves: The AUCs for FXIII, platelet count and fibrinogen concentration were significantly different from each other, indicating that FXIII activity was a significantly better predictor of red cell transfusion than platelet count and fibrinogen concentration.

**Table 1 jcm-09-02456-t001:** Prevalence of overall factor XIII (FXIII) deficiency, fibrinogen deficiency, and thrombocytopenia.

FXIII Activity (%)	Fibrinogen (g/L)	Platelets (G/L)	*n*	Proportion
<70	any	any	515	50%
<60	any	any	399	39%
any	<1.5	any	93	9%
any	<1.0	any	25	2%
any	any	<150	503	49%
any	any	<100	249	24%

**Table 2 jcm-09-02456-t002:** Prevalence of combined deficiencies.

	FXIII Activity (%)		Fibrinogen (g/L)		Platelets (G/L)	*n*	Proportion
A	≥70	and	≥1.5	and	≥150	302	30%
B	**<70**	and	≥1.5	and	**<150**	229	22%
C	**<70**	and	≥1.5	and	≥150	204	20%
D	≥70	and	≥1.5	and	**<150**	195	19%
E	**<70**	and	**<1.5**	and	**<150**	72	7%
F	**<70**	and	**<1.5**	and	≥150	10	<1%
G	≥70	and	**<1.5**	and	**<150**	7	<1%
H	≥70	and	**<1.5**	and	≥150	4	<1%

Median numbers of transfused red blood cell (RBC) concentrates in patients with the respective abnormalities were two (range 0–12) in patients with low FXIII but normal platelet count and normal fibrinogen (*n* = 204, group 1); zero (range 0–18) for low platelet count and normal FXIII and fibrinogen (*n* = 195, group 2); and zero (range 0–0) for low fibrinogen with normal FXIII and platelet count (*n* = 4, group 3). Number of transfused RBCs were significantly different between groups 1 and 2 (*p* = 0.0039) and groups 1 and 3 (*p* = 0.036); the difference between groups 2 and 3 was not significant (*p* = 0.13), possibly due to the small patient number in group 3. Bold caption marks lines where proportion of results below the given reference ranges are reported.

**Table 3 jcm-09-02456-t003:** Odds ratios for receiving red blood cells, according to a respective deficiency/combination of deficiencies.

Variables Evaluated	OR	OR 95% CI
FXIII activity < 70%	**4.62**	**3.49–6.11**
Platelet count < 150 G/L	**1.96**	**1.48–2.59**
Fibrinogen < 1.5 g/L	1.08	0.67–1.76
FXIII < 70% AND platelets < 150 G/L	**3.18**	**2.35–4.31**
FXIII < 70% AND fibrinogen < 1.5 g/L	2.51	0.89–7.07
Platelets < 150 G/L AND fibrinogen < 1.5 g/L	0.60	0.20–1.76

Statistically significant results are bolded. OR = Odds Ratio; CI = Confidence Interval.

**Table 4 jcm-09-02456-t004:** Receiver operating characteristics (ROC) curves’ areas under the curve (AUC) of FXIII activity, platelet count, and fibrinogen for the prediction of red cell transfusion.

Variables Evaluated	AUC	AUC 95% CI
FXIII activity < 70%	0.744	0.716–0.770
Platelet Count < 150 G/L	0.632	0.601–0.661
Fibrinogen Concentration < 1.5 g/L	0.578	0.547–0.609

CI = Confidence Interval.

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
