# Peer review of "The Need for Red Cell Support During Non-Cardiac Surgery Is Associated to Pre-Transfusion Levels of FXIII and the Platelet Count"

_jcm, 2020, doi:10.3390/jcm9082456_

Round 1
Reviewer 1 Report
The authors have appropriately responded to the comments proposed and improved the manuscript by making the corrections/modifications.
On re-review of this manuscript, one important consideration came into question about the timing of the blood samples being drawn and some of the discussion (lines 232-235). It seems the blood samples were taken after the patient hand intraoperative bleeding that prompted the clinical decision to give blood transfusion. Therefore, there already was consumptive loss of the Factor XIII/plts/fibrinogen. This is subtle but important point and so the discussion should clearly state that the levels measured are not measured PRE-operatively, i.e. before the patient even has surgery. While it is interesting and informative that Factor XIII levels can predict transfusion once a patient has bleeding, it would be even more interesting if the presurgical specimens showed a correlation to predict transfusion as well.
Reviewer 2 Report
Changes made to the manuscript improved clarity and readability. Great work.
Author Response
Dear Reviewer
Again we are very thankful for reading our manuscript and your critical assessment of our work. We are glad to notice, that the revised version seemed to be suitable from your point of view.
This manuscript is a resubmission of an earlier submission. The following is a list of the peer review reports and author responses from that submission.
Round 1
Reviewer 1 Report
Listyo et al. presents an interesting and compelling analysis on the role that Factor XIII may play in pre-operative risk prediction for transfusion during delivery. Despite being retrospective in nature, the sample size is impressive and the results striking. I have additional suggestions/comments that can strengthen this manuscript further:
1) Overall comment: Throughout the entire manuscript it was a bit unclear the study population that was studied. For example, the title, abstract and introduction mention very little that the population was pregnant women at delivery but rather focus on 'non-cardiac'. I think it is important to be clear, particularly for the OB/Gyn reader and in the abstract since this was not mentioned anywhere, that the study group was pregnant women at delivery.
2) Introduction: I would suggest being more specific about the % prevalence of Factor XIII deficiency after surgery, which I believe is attributed to citation 21 though not entirely clear. This is important to have context about how often this happens in other surgical fields to then be able to compare to the findings presented in this paper.
3) Introduction: Related to comment #1, I would suggest that at the end of the introduction to make this clear that tit was the 'routine obstetric surgical setting'.
4) Methods: Please clarify what years were the samples collected from. Please also clarify if all blood tests were after delivery and prior to blood transfusion or were some samples prior to delivery on patients who were high risk for needing transfusion. This is important since the delivery itself will cause severe bleeding and may compound coagulopathy.
5) Results: It would be interesting if the authors provided additional AUC statistics just in the text of the combined abilities for these components: FXIII+fibrinongen, FXIII+platelets, fibrinogen+platelets. I realize this would make the figure much more difficult to read if these were included on it so just the analysis with statistics would be meaningful.
6) Discussion primarily: Please correct terms such as 'prepartal' to say antepartum and 'postpartal' as postpartum. These are the most commonly used English terms for predelivery and postdelivery. Similarly, there are some minor other choices of English words such as 'retard' transfusion in the maternal and methods that is not universally understood and instead 'avoid' may be more appropriate.
7) Discussion: Lines 183-189 could use additional clarity by reducing how many parentheses are used since this makes the main point of these two sentences unclear.
8) Conclusion: I am not sure I would agree further studies are needed to see if platelet transfusion is needed since many women with a platelet count between 100-150K often do fine during delivery and so I would not want to suggest that this may be beneficial. However, a threshold definition may be helpful since often <50K is cited often as the point below which it would be dangerous at delivery but limited data guide specific transfusion of platelets. I think Factor XIII replacement may be interesting but would be helpful to have some information in the discussion if this is currently given for any other surgeries either for prophylaxis or treatment? Just in Europe or worldwide?
Reviewer 2 Report
Brief summary:
This research paper adresses an important question: what are the predictors of intra operative bleeding risk, more specifically coagulation laboratory markers, and therefore risk of transfusion during surgery. Understanding the pathophysiology of operative bleeding is very relevant to subsequently be able to hold trials that address prophylactic interventions, and even influence the choice of blood products to transfuse to specifically tackle the right form of coagulopathy. Following previous findings that FXIII and thrombocytopenia were frequent modulators of clot firmness in the post-operative setting, it is a natural next step to examine the relationship between these laboratory markers before the bleeding event and the outcome of transfusion. The investigators did a great use of data available to them, namely routine FXIII determinations, that are not available in all centres. FXIII holds a key role in clot formation, yet is often overlooked in standard laboratory testing, and this research adds to the bulk of evidence that low FXIII increases the risk of bleeding in surgery with significant results. The limits of the study are also clearly stated (retrospective, single center) by the authors.
Broad comments:
1) Population studied: It would be interesting to know what type of elective non-cardiac surgeries were performed in this cohort to help the reader better identify the population to which the results of this study can apply.
Also, how was the cohort identified for the look back analysis? (surgery lists, blood bank system, archives, …)
2) Methods: The methods would benefit from further clarification as it is not clear from the first reading how the cohort was chosen (as described above) and what the comparator arm was to determine the odds ratio of receiving red blood cell (RBC) products. Examples in writing that could contribute to create confusion are:
Line 17 (abstract): «… 1023 patients, which were in need of transfusion in the operating theatre. »
Line 78-79: « This was combined with a look back on red blood cell concentrates delivered to the operating room during the same period. » Does this include the blood products that were delivered but not transfused? Perhaps specify.
Line 82-84: «… all patients had routine platelet count, fibrinogen and FXIII determination drawn in the operating room immediately before red cell transfusion was given. »
Line 104-106: « …, data sets of 1023 patients, who had received blood products, were identified for further evaluation. »
These statements could imply that all patients were transfused in the OR, which obviously was not the case when looking at the data. This leaves us wondering: did all patients receive transfusion during the hospitalization and the odds ratio represents risk of being transfused during surgery versus later? Or is the odds ratio really a risk of transfusion of RBC versus no transfusion? Or is it a risk of receiving RBC versus other blood products?
3) Results: The proportion in the text and various tables needs to be better defined. The presentation of results would also benefit if the number (or proportion) of patients transfused with RBC products among the patients who had a single or a combined deficiency was stated.
Minor comments:
Line 25-26: « All AUCs were significantly different from each other (p<0.0001 and p=0.0106, respectively. » While we understand from text (line 147-149) that this means that FXIII levels was a significantly better predictor of RBC transfusions, the abstract would read better if this point was made clearly instead of focusing on the result that AUCs were significantly different from each other (that could be different in any direction including FXIII being the worst predictor.)
Line 37: The statement that transfusion (as a treatment of blood loss) is associated with significant morbidity is widely recognized but would be better supported by a contemporary reference as transfusion associated infectious risk has changed over the last decades (concerning Reference 3 from NEJM 1992). A reference encompassing broader transfusion associated adverse events (more than just infectious risks) would also be better suited.
Line 62-63: I doubt that reference 19 (a pharmacodynamics animal study) supports well the statement that «The use of FXIII led to a significant reduction of fibrinogen consumption, blood loss and loss of clot firmness. »
Line 106: Perhaps specify in which patients the 1636 RBC units were transfused (in the entire cohort of 1023 or in the 443 who were transfused in the OR).
